# The Mechanism of Hyperglycemia-Induced Renal Cell Injury in Diabetic Nephropathy Disease: An Update

**DOI:** 10.3390/life13020539

**Published:** 2023-02-15

**Authors:** Tong Wu, Lei Ding, Vivian Andoh, Jiaxin Zhang, Liang Chen

**Affiliations:** 1School of Life Sciences, Jiangsu University, Zhenjiang 212013, China; 2School of Food Science and Biological Engineering, Jiangsu University, Zhenjiang 212013, China

**Keywords:** diabetic nephropathy, hyperglycemia, mechanism, renal cell injury

## Abstract

Diabetic Nephropathy (DN) is a serious complication of type I and II diabetes. It develops from the initial microproteinuria to end-stage renal failure. The main initiator for DN is chronic hyperglycemia. Hyperglycemia (HG) can stimulate the resident and non-resident renal cells to produce humoral mediators and cytokines that can lead to functional and phenotypic changes in renal cells and tissues, interference with cell growth, interacting proteins, advanced glycation end products (AGEs), etc., ultimately resulting in glomerular and tubular damage and the onset of kidney disease. Therefore, poor blood glucose control is a particularly important risk factor for the development of DN. In this paper, the types and mechanisms of DN cell damage are classified and summarized by reviewing the related literature concerning the effect of hyperglycemia on the development of DN. At the cellular level, we summarize the mechanisms and effects of renal damage by hyperglycemia. This is expected to provide therapeutic ideas and inspiration for further studies on the treatment of patients with DN.

## 1. Introduction

DN is a common complication of diabetes mellitus in which the structure and function of the kidneys are disturbed. It is also considered a glomerular disease. Persistent urinary protein (albumin excretion rate (AER) > 300 mg/24 h) in diabetic patients without other renal disease and urinary tract infection is defined as DN [1]. The pro-inflammatory and profibrotic effects lead to the apoptosis of mesangial and endothelial cells and, ultimately, to the disruption of the glomerular filtration barrier. Abnormalities in glomerular and tubular function occur almost simultaneously [2]. Hypertension, progressive proteinuria, glomerulosclerosis, and decreased glomerular filtration rate (GFR) are the main features of DN [3]. Poor blood glucose control in patients with chronic diabetes aggravates these major pathological features and produces structural changes in the kidney. The most common pathological changes in the glomerulus are telangiectasia, basement membrane thickness, extracellular matrix expansion, multicellular injury, and fibrosis [4]. According to the ultrastructural changes of glomeruli, the glomerular lesions of DN are divided into four stages: Grade I, thickening of glomerular basement membrane (GBM), which is characterized by ultrafiltration; Grade II, mild mesangial dilatation (IIa) or severe Mesangial dilatation (IIb); Grade III, tuberous sclerosis (Kimmelstiel-Wilson lesion); and Grade IV, advanced diabetic glomerulosclerosis with obvious albuminuria [5]. GBM occurs in the early stage of diabetes, and the extracellular matrix accumulates over time; the mesangial matrix thickens continuously, resulting in the diffuse expansion of the mesangial matrix, which is considered a landmark lesion of DN [6]. Diabetes affects all cell types of the kidney, including endothelial cells, tubular interstitial cells, podocytes, and mesangial cells [7]. The effects of high glucose, polyol pathway activation, hexosamine pathway activation, and other mechanisms lead to structural and functional abnormalities in various renal cells [8,9].

Many risk factors affect DN, such as heredity, hyperglycemia, hypertension, obesity, and lifestyle [10,11]. Some researchers have confirmed that the occurrence of hyperglycemia promotes glomerular hyperfiltration. At the same time, hypertension increases the incidence of proteinuria and focal segmental glomerulopathy, both of which cause damage to the kidneys [12]. In type 1 diabetics, insufficient insulin secretion reduces the ability to store glucose in muscle and adipose tissue, or otherwise in the liver, thus exacerbating chronic hyperglycemia [13,14]. Insulin resistance in people with type 2 diabetes reduces glucose uptake by skeletal muscle, and insulin resistance in the liver increases endogenous glucose production, exacerbating the toxic effects of glucose [15]. It follows that glucose homeostasis disruption characterizes type 1 and type 2 diabetes. The kidneys are increasingly recognized as an important regulator of glucose stability. This is because the kidneys actively reabsorb filtered glucose into the bloodstream and produce glucose through gluconeogenesis. It is estimated that the kidneys are responsible for approximately 20–25% of systemic glucose release and 40% of all gluconeogenesis in the body [16,17]. The prolonged exposure of kidney cells to a glucose-rich environment exacerbates the extent of damage, triggers the polyol pathway, activates protein kinase C (PKC), enhances the hexosamine biosynthesis pathway (HBP), promotes the formation of AGEs, and promotes DN formation [18]. In diabetes patients, hyperglycemia can induce the excessive formation of the extracellular matrix and glomerular mesangial dilatation. It then blocks glomerular capillaries and gradually affects the normal physiological activity of the kidney [19]. It has been demonstrated that blood glucose control positively affects the progression of renal disease. Thus, improving blood glucose control and intensive insulin therapy can reduce glomerular tissue deterioration in patients with type 1 diabetes, and the effect of blood glucose on kidney disease has the same conclusion in patients with type 2 diabetes [1]. Therefore, summarizing and exploring the microenvironmental imbalance and diffuse cellular abnormalities disturbed by hyperglycemia is significant for the progressive structural changes and clinical diagnosis of DN.

This review focuses on the pathogenesis associated with hyperglycemia-induced renal injury and the mechanisms of damage to major renal cells. It is clear that DN is an early metabolic event triggered in a hyperglycemic environment, during which the accumulation of various by-products, dysregulation of signaling pathways, and induction or initiation of certain cellular pathological processes, such as alterations in cell cycle and growth, produce toxic effects on various cells in the kidney. Elaborating on the mechanisms of chronic hyperglycemia damage to DN and irreversible renal cell damage provides new perspectives for the future understanding and treatment of DN.

## 2. Key Regulators of Hyperglycemia-Mediated Cell Injury in DN

Chronic hyperglycemia is one of the leading causes of DN. The control of blood glucose and blood pressure has always been the treatment and management strategy for patients with DN. Although the harm of hyperglycemia to patients has been widely recognized, not all transient hyperglycemia is harmful. Transient self-limited hyperglycemia is considered beneficial to the host during periods of non-severe disease in diabetic patients [20]. Fasting hyperglycemia occurs in almost all patients with diabetes, which may be due to hormonal imbalance, resulting in increased liver glucose transport. Increased glucose transport means an imbalance in glucose homeostasis, such as decreased sensitivity of the liver to insulin or decreased insulin, resulting in increased hepatic glucose production and decreased peripheral glucose utilization [21]. When the blood glucose level is normal, most glucose will be metabolized to pyruvate through glycolysis under aerobic conditions. Pyruvate is oxidized and decarboxylated by pyruvate dehydrogenase complex to acetyl CoA, which enters the tricarboxylic acid cycle. Finally, it oxidizes to water and carbon dioxide, releasing energy for the body to use. However, in the high glucose environment, the glycolysis process is saturated. Most glucose cannot complete the normal metabolism, which stimulates the collateral metabolism pathways, such as the polyol and sorbitol pathways. The excessive activity of collateral metabolism leads to metabolic imbalance, causing biochemical disorders in key tissues (including the kidney), resulting in irreversible changes in function and even structure [22]. In DN, the main cellular targets of hyperglycemia injury are glomerular endothelial cells (GECs), mesangial cells (MC), podocytes, and endothelial cells. Among them, highly specialized podocytes are vulnerable to apoptosis, which increases glomerular permeability, so DN is sometimes referred to as a “podocyte centrality” disease [23,24].

There is a non-negligible relationship between the drivers of multicellular damage and hyperglycemia. Previous studies have shown that high glucose can activate the accumulation of advanced glycation end products (AGEs) are also involved in the pathogenesis of DN [25]. There are f main hypotheses about how hyperglycemia causes diabetic complications (including DN), which are the polyol pathway, hexosamine pathway (HBP), production of AGEs, and PKC, and they have been widely supported by scholars. Although these four different mechanisms interpret the injury process induced by hyperglycemia from different angles, they are not entirely independent. For example, the mitochondrial electron transport chain produces excess superoxides, which are cleverly linked as a common factor of the four mechanisms [26,27]. There are many more such connections, but they are not the focus of this paper and, therefore, will not be elaborated on. In the subsequent write-up, we will describe the mechanisms of renal damage from hyperglycemia and try to summarize the four types of renal cells affected by the high glucose environment. Figure 1 shows the route of this paper.

### 2.1. Polyol Pathway

The polyol pathway is the primary collateral pathway in glucose metabolism, which mainly includes two steps: first, the reduction of glucose to sorbitol by nicotinamide adenine dinucleotide phosphate (NADPH) under the action of Aldose reductase (AR), and the second, the sorbitol dehydrogenase (SDH) uses the NAD^+^ cofactor to catalyze the oxidation of sorbitol to fructose [22]. AR and SDH are two key enzymes in the polyol pathway. Sorbitol is a strong hydrophilic alcohol with polyhydroxyl groups that cannot pass through the cell membrane. The excessive activation of the polyol pathway leads to the excessive accumulation of sorbitol in cells, producing cellular osmotic regulatory substances to balance the extracellular hyperosmotic stress [28,29]. This may be the reason why the polyol pathway maintains the extracellular and intracellular osmotic pressure balance during changes in glucose concentration. During the period of high glucose, the polyol pathway is also a major cause of oxidative stress induced in diabetes based on three points: (1) Fructose produced by the polyol pathway can be phosphorylated to fructose-3-phosphate, which is then decomposed into 3-deoxyglucosone. Fructose-3-phosphate and 3-deoxyglucosone are strong glycosylation reagents that accelerate the formation of the AGEs [30]. The binding of AGEs and AGEs to their receptors will lead to oxidative stress. (2) The increase in AR activity accelerates the consumption of NADPH. It inhibits the activity of glutathione reductase, resulting in a decrease in the synthesis of reduced glutathione (GSH), which reduces the ability of cells to respond to oxidative stress [31]. (3) During the bioconversion of SDH from sorbitol to fructose, the cofactor NAD^+^ of SDH will be transformed into NADH, and excessive NADH may become the substrate of NADH oxidase, which may accelerate the production of ROS [32]. Overactive polyol pathways disrupt the antioxidant capacity of cells and abnormal oxidative stress response, which in turn accelerates the awakening of multiple cellular damage mechanisms, leading to the development of multiple complications, including DN. At the same time, the activity of the polyol pathway does not occur in isolation. Some researchers have found that AR can drive the pentose phosphate pathway while inducing NADPH and finally stimulate protein kinase C to induce glomerular prostaglandin accumulation and decrease the contractility of glomerular Mesangial cells, resulting in glomerular dysfunction [33].

### 2.2. Hexosamine Pathway

The hexosamine biosynthesis pathway (HBP) is a glucose metabolic pathway. Glucose is converted to glucose-6-phosphate by hexokinase and then to fructose-6-phosphate (F-6-P) as part of the glycolysis pathway. Hexokinase is an important enzyme involved in the transport of glucose to cells. The expression of this enzyme is controlled by the glucose-6-phosphate dehydrogenase (G6PDH) [27,34]. In the hyperglycemic environment of diabetes, most F-6-p is converted to glucosamine-6-phosphate (GlcN6-P) under the action of F-6-p aminotransferase (GFAT), which is eventually metabolized into various aminohexose products, including uridine diphosphate N-acetylglucosamine (UDP-GlcNAc) [35]. HBP may play its role through two mechanisms: (1) UDP-GlcNAc is the precursor of glycoprotein, proteoglycan, glycosaminoglycan, and all other amino sugars [36], and (2) when glucose concentration is high, glycolysis gradually converts fructose-6-phosphate in HBP to UDPN-acetylglucosamine. At the same time, N-acetylglucosamine modifies cytoplasmic and nuclear proteins through mono-O-linked N-acetylglucosamine (O-GlcNAc) glycosyl groups and affects their functions [8]. HBP is involved in mediating high glucose levels, and it is the mediator of the effect of high glucose on the accumulation of the Mesangial and renal tubular matrix. Previous reports have shown that HBP is involved in hyperglycemia-induced production of transforming growth factor (TGF-β1), with the rate-limiting enzyme GFAT, and that TGF-β1 mediates hyperglycemia-induced mesangial and tubular matrix formation, further accelerating the process of nephrosclerosis in DN [37]. In addition, some scholars have found that O-GlcNAc may contribute to the harmful effects of high glucose in mesangial cells: O-GlcNAc in HBP can activate p38 mitogen-activated protein kinase (p38MAPK) in glomerular membrane cells to stimulate the activation of the fibrosis signal pathway and control the expression of plasminogen activator inhibitor-1, fibronectin, and transforming growth factor β, which can cause DN matrix accumulation [38]. The increase of O-GlcNAc modification of protein in renal biopsy specimens in diabetes mellitus, accompanied by diffuse nuclear and cellular solute staining; it also proved the negative effect of O-GlcNAc on the kidney [39]. It can be seen from the above that there seems to be a close relationship between HBP-mediated DN kidney injury and the two mechanisms mentioned above. Still, the molecular law and structural relationship between the two mechanisms must be further explored.

### 2.3. Production of Advanced Glycation End Products (AGEs)

Glycosylation occurs when reducing sugars (such as glucose) undergo a series of non-enzymatic reactions with ketones or aldehydes, proteins, lipids, and nucleic acids to form Schiff bases. Reversible Schiff bases compounds are transformed into more stable covalently bound Amadori products, which are further rearranged to form irreversible heterochemicals called advanced glycation end products (AGEs), a process also known as the Maillard reaction [40]. In the Maillard reaction, Amadori products produce intermediate compounds called a-dicarbonyl soroxo aldehydes when they undergo rearrangement, which is formed by non-oxidative rearrangement and hydrolysis of fructose-3 phosphoric acid and Amadori. It is also considered to be one of the reasons hyperglycemia aggravates diabetes and related complications [41]. Pentosidine is a catabolite of the AGE structure. The kidney plays a central role in removing free pentosidine. Free pentosidine is filtered through the glomerulus and then reabsorbed in the renal tubules, and it is modified or degraded in the renal tubules. The primary lesions of renal tubular cells, early lysosomal dysfunction, cause the filtered pentose glycosides to be largely reabsorbed by the renal tubules and ultimately removed from the urine [42]. Increased AGE-free adducts were found in renal failure. AGE-free adducts are the main form of AGEs clearance from the body, and reduced renal clearance is the main cause of this phenomenon [43]. There is also a causal relationship between the formation of AGEs and the pathological changes of DN. Increased AGEs in patients with DN can exacerbate the condition of patients with DN. It has been reported that tissue AGEs levels in patients with end-stage DN are almost twice as high as those in patients with diabetes without renal complications [44].

In patients with DN, the increase in AGEs formation may be related to the following: (1) the organs of DN patients, including kidneys, were exposed to a high glucose environment for a long time, which increases the probability of glycosylation and accumulation of AGEs. AGEs activate different signal pathways, such as mitogen-activated protein kinase/extracellular regulated protein kinase (MAPK/ERK), induce oxidative stress, endoplasmic reticulum stress inflammation and fibrosis, and accelerate renal pathological changes [45]. This fully shows that there is a close relationship between hyperglycemia and AGEs. (2) RAGE is the receptor of AGEs. The expression of RAGE protein in podocytes and mesangial cells of DN patients increased, which activated AGE-RAGE signal transduction and accelerated the accumulation of AGE and glomerular injury [46]. The reduction in AGEs clearance may be related to the following: (1) the interstitial injury results in low filtration of renal tubules, which is reabsorbed into the blood by proximal renal tubules [42]. (2) AGEs also decreased the degradation of AGEs by matrix metalloproteinases, resulting in the thickening of the basement membrane [47]. (3) AGEs can induce Mesangial cell apoptosis, stimulate the secretion of VEGF (vascular endothelial growth factor) and MCP-1 (monocyte chemoattractant protein-1), reduce glomerular filtration rate, and thus participate in DN lesions [48]. In view of the promoting effect of hyperglycemia on the accumulation of AGEs, controlling blood glucose and striving to remove or degrade AGEs is expected to improve the pathological state of DN and delay the disease in DN.

### 2.4. Protein Kinase C

Protein kinase C (PKC) is a family of enzymes that can be activated by the second messengers, such as calcium (Ca^2+^), diacylglycerol (DAG), and other lipid physiology [49]. A variety of isomers are distributed in many cell types located in the cytoplasmic part at rest. The activation of PKC is usually associated with the redistribution of enzymes from the cytoplasmic part to other sites [50,51]. Hyperglycemia can activate DNG, and DAG is closely related to PKC. The increase in blood glucose level may increase the production of DAG through several metabolic pathways, such as dihydroxyacetone phosphate reduction to glycerol 3-phosphate during glycolysis and gradual acylation to increase DAG. The activity of phospholipase D and glycolytic enzyme glyceraldehyde-35 phosphate dehydrogenase also regulates the synthesis of DAG [27,52]. Diacylglycerol (DAG)-PKC pathway (DAG-PKC) is a potential mechanism leading to diabetic microvascular complications. PKC is a sensor with a rapid rise of DAG. Calcium-dependent PKC can regulate the membrane localization function according to the presence of DAG in the membrane and complete the rapid movement from the cytoplasm to the membrane and back [53]. With the transient increase of plasma membrane DAG, DAG and its main target PKC also regulate other important cellular responses.

Some scholars have found that increasing glucose concentration can inhibit the phagocytosis mediated by the granulocyte complement receptor and FC γ receptor in a dose-dependent manner. At the same time, inhibition of PKC can reverse this process [54]. Hyperglycemia-activated PKC can also mediate and participate in other mechanisms. It is well known that extracellular matrix (ECM) accumulation is one of the glomerular lesions of DN. The collection of ECM is related to transforming growth factor (TGF-β1) [55,56]. Hyperglycemia can increase the activity of PKC, and the activated PKC can increase the expression of TGF-β1, resulting in an increase in glomerular ECM production [57]. The activation of PKC induced by glucose concentration may also mediate the inhibitory effect of glucose on insulin-stimulated chemokines and the production of macrophage factors, such as interleukin-1 (IL-1) [58,59]. This finding suggests that glucose-induced PKC activation mediates the phagocytosis of granulocytes. It indicated that glucose-induced PKC activation was involved in renal injury.

## 3. Renal Cell Types and Mechanisms Associated with Hyperglycemia-Induced Injury in DN

Complex mechanisms and renal cell abnormalities contribute to the changes of characteristic tissues in the kidney of DN, which become the pathological detection index and evaluation basis of clinical DN. The damage to renal cells is dynamic, compound, and connected. Multicellular damage affects other cells, showing structural and functional abnormalities in tissues and organs. Although a complete blueprint on the multiple renal cellular injuries in DN has not been clearly summarized so far by attempting to learn and summarise the typical cell types and possible mechanisms of high glucose injury, it will help better to understand the pathogenesis of DN at the cellular level. It will be helpful for cell biology studies of DN. On the basis of the available literature, we have attempted to summarise the mechanisms of damage to the four types of renal cells. The main mechanisms mentioned in the paper are illustrated in Figure 2, Table 1.

### 3.1. Mesangial Cell

Glomerular mesangial cells (MC) are the contractile cells that make up mesangium and support glomerular villi structurally. Glomerular mesangial cells are complex reticular structures composed of extracellular matrix proteins, including type IV collagen, laminin, fibronectin, and proteoglycan. Podocytes and endothelial cells secrete the components of glomerular mesangial cells to form a hybrid basement membrane [60]. Hyperglycemia has been shown to cause kidney damage in patients with diabetes [61]. Studies have shown that high glucose concentration can reduce Mesangial cells’ ability to degrade the Mesangial matrix by affecting the activity of matrix metalloproteinases (MMPs) [62].

The effect of high glucose in blood on the function and structure of the glomerular. Mesangial cells are mainly reflected in the following aspects. (1) High glucose affects the function and formation of mesangial-associated proteins; when the concentration of glucose reaches a certain concentration, it will induce Mesangial cells to produce endogenous glucose transporter-β 1 (GLUT-β 1) and then induce the overexpression of GLUT-1 mRNA and produce a large amount of GLUT-1 protein. GLUT-1 protein can promote glucose transport and stimulate glucose uptake. Exposure to MC in a high glucose environment induces gene expression and the protein secretion of collagen, laminin, and fibrin, which is one of the factors responsible for extracellular matrix accumulation (ECM) [63]. (2) High glucose affects mesangial proliferation; high glucose induces the early activation of the PDGF loop, PDGF-B chain gene expression, and continuous increase, resulting in increased TGF-β1 gene expression, thus regulating the proliferation of Mesangial cells and the production of the Mesangial matrix [64]. Some researchers also found that after long-term exposure to high glucose MC, the cell cycle stagnated in the G1 phase, and the cells showed obvious proliferation and hypertrophy [65]. (3) For high glucose-induced glomerular Mesangial cell apoptosis, the apoptosis of glomerular Mesangial cells induced by high glucose is mediated by the Wnt/β-catenin signal pathway. High glucose can down-regulate the expression of Wnt4 and Wnt5a and the nuclear translocation of β-catenin while increasing glycogen synthase kinase-3 β (GSK-3 β) and caspase-3 and apoptosis of glomerular Mesangial cells [66]. Similarly, hyperglycemia can also increase the apoptosis rate of mesangial cells by activating the intrinsic pathway of proapoptotic signal transduction in mesangial cells or enhancing the sensitivity of endogenous TGF-β1 [67,68]. (4) For high glucose-induced cytokine changes, high glucose changes cytokine environment will indirectly or directly affect the MC response to internal environment metabolism. High glucose can induce the TGF-β type II receptor [69], excessive expression of connective tissue growth factor (CTGF) [70], increased production of circulating angiotensin II (Ang II) [71], and increased release and recruitment of monocyte chemoattractant protein-1 (MCP-1) [72] produced by MC pathological changes, such as DN glomerular hypertrophy, sclerosis, and ECM deposition. (5) For high glucose changes related to metabolism and signal transduction, high glucose inhibits the glycolysis pathway of MC by inducing the production of reactive oxygen species (ROS) through the electron transport chain and inhibiting the activity of glyceraldehyde-3-phosphate dehydrogenase (GAPDH) [73]. Some researchers have identified polyol pathway enzymes in MC and found sorbitol accumulation in MC under high glucose conditions [74], activation of the RAGE signal pathway [75], activation of protein kinase (PKC) [76], and so on.

Although the various mechanisms of the effects of high glucose on MC briefly summarized above are summarized separately, they do not represent the complete independence and communication between the mechanisms. It affects the cell itself and its communication and function with other cells around it, which plays an important role in glomerular and tubular injury.

### 3.2. Glomerular Endothelial Cell

There are a variety of endothelial cells in the kidney, such as glomerular endothelial cells (GECs), microvascular endothelial cells of the capillaries around the renal tubules, and larger venous and arterial endothelial cells. These cells are distributed in different locations and environments of the kidney and participate in various functions. GECs are in direct contact with blood, are closely distributed on the glomerular capillary lumen surface, and share a glomerular basement membrane (GBM) with podocytes. Together, they form an interconnected glomerular filtration barrier [77]. However, the complete GECs barrier includes glycocalyx and endothelial surface layer (ESL), cell–cell junction, and subendothelial glycocalyx and matrix that cover and fill the window. Hyperglycemia aggravates filtration, and large windowing has the lowest contribution to albumin retention without bridging the outer septum [77,78]. The pathological changes of GECs and podocytes will aggravate the dysfunction of the glomerular filtration barrier while the phenotype of GECs will change according to the surrounding environment. In the process of endothelial-interstitial transformation (EMT), the conversion of GECs to interstitial will accelerate renal fibrosis and the development of the DN [79]. Glomerular extracellular matrix protein (ECM) deposition is responsible for renal fibrosis, and its effector cells are thought to be myofibroblasts. Previous studies have shown a causal relationship between EMT and myofibroblasts [80] and that myofibroblasts originate from the epithelial cells of the EMT [81]. This was reinforced by Iwano et al., who showed that up to 36% of myofibroblasts could be produced through local EMT of renal tubular epithelial cells during renal fibrosis [82].

In addition, Endothelin-1 (ET-1) is an active polypeptide composed of various amino acids, which exists widely in various tissues and cells. Hyperglycemia can also induce GECs to produce ET-1 while ET-1 is regulated by HIF-1α, and ET-1 can mediate EMT in GECs caused by high glucose [83]. In experimental DN, the increase in serum ET-1 concentration was related to increased urinary albumin and NAG levels, which damaged diabetic lesions [84]. Glomerular endothelial mitochondrial dysfunction was also associated with the increased expression of glomerular ET-1 receptor An and increased circulating ET-1. Meanwhile, glomerular ET-1 receptor-induced mitochondrial oxidative stress and dysfunction, resulting in podocyte depletion, which was alleviated or did not even occur when dysfunctional endothelial cells were treated with mitochondrial antioxidant mito-TEMPO [85]. Furthermore, hyperglycemia can also independently lead to endothelial dysfunction, which is associated with decreased production or bioavailability of endothelial-derived substances, such as nitric oxide (NO), which can induce loss of vascular smooth muscle relaxation, platelet aggregation, and reduced leucocyte adhesion. In addition, this glucose-mediated NO loss can lead to hypertension. The physical damage caused by hypertension (such as increased shear stress and blood flow-related force) will further aggravate renal microvascular damage [86]. Although the effects of hyperglycemia-induced endothelial injury and its related disturbances in DN are undeniable, the complex structure and molecular interaction of endothelial barrier function remain unknown. The immediate explanation for the increased permeability due to barrier dysfunction is the loss of glycocalyx and the shedding of glycosaminoglycans in the window area [87]. Various hypotheses and conjectures are gradually being explored and proved, which further indicates that the biological process of glucose-induced GECs dysfunction plays an essential role in the development of DN.

### 3.3. Podocyte

Podocyte structure is a complex, epithelial cell layer composed of the cell body, microtubule-rich main protrusions, and actin-based foot processes. Podocytes tightly enclose and support the glomerular capillaries, and the cell body extends many protrusions, cross-finger-like, attached to the outside of the GBM. The selective function of glomerular filtration is mainly the gap membrane proteins between foot processes (such as nephrin and podocin) and ensures contact between podocytes [88]. DN patients with poor control of blood glucose and blood pressure, fissure diaphragm (SD) protein destruction, and endothelial dysfunction is often accompanied by podocyte disappearance (Effacement) [89].

High glucose induces podocyte injury, and podocytes undergo EMT after the loss of epithelial markers (such as Nephrin, P-cadherin, and atresia zone-1). Injury is induced by mesenchymal markers (including nephrin protein, fibroblast-specific protein-1, and matrix), ultimately leading to renal tubule injury and interstitial fibrosis [90,91]. It is worth mentioning that nephrin is located in the podocyte slit diaphragm and belongs to the immunoglobulin superfamily of cell adhesion molecules. It also ensures that the ultrafiltration process of the mediastinum is used to maintain the integrity of the structure. Snail is a kind of DN binding molecule that can regulate cell adhesion and is one of the key transcription factors that initiate EMT and regulates nephrin [92]. High glucose can directly or indirectly induce EMT in podocytes, increase the level of glucose, and up-regulate the protein expression of Snail, thus inhibiting the protein expression of P-cadherin and nephrin [93], resulting in podocyte depletion. Matsui et al. also found that GSK3 regulated the expression level of Snail in injured podocytes, and the activity of GSK3 decreased in injured podocytes, which partly led to the decrease of nephrin and the increase of Snail [92], resulting in damage in podocytes. In addition, high glucose increased the expression of α-SMA and desmin proteins by triggering the activation of the PI3K/AKT signal pathway in podocytes. The protein expression of podocalyxin and nephrin was inhibited [94], which could also induce the phenotypic transformation of podocytes. The analysis of podocyte energy metabolism suggests that disturbed energy conversion may be an additional cause of induced podocyte damage. The sensitivity of podocytes to oxidative metabolism may be due to the abundant mitochondria in the peripheral foot process [95]. It has also been found that the high glucose environment interrupts oxidative phosphorylation, glycolytic conversion, and abnormalities in downstream signaling networks in podocytes, inhibiting the production of related factors, such as reduced expression of myocyte-specific enhancer 2C (MEF2C), myogenic factor 5 (MYF5), and receptor-γ cofactor 1a (PGC-1α) [96].

In addition, hyperglycemia can directly produce toxicity to cells, which can lead to the stress response of podocytes, resulting in cell cycle arrest, hypertrophy, shedding, and apoptosis. Some researchers have found that the CARM1-AMPK α-Notch1-CB1R signal mediates podocyte apoptosis induced by high glucose, and glucose-induced down-regulation of CARM1 is due to ubiquitin-dependent CARM1 degradation [97]. Loss of the foot process (FPE) is also related to the structure and function of podocytes, such as cytoskeleton rearrangement, apoptosis, and autophagy damage [98]. Experimental studies have shown that podocyte hypertrophy is related to the pathological development of progressive DN, such as promoting glomerular hyperfiltration [99], and hyperglycemia can promote this process. Some studies have shown that hyperglycemia can activate the upstream element Gp130 to induce nuclear STAT3 expression and cause podocyte hypertrophy [100]. Thus, hyperglycemia damage to podocytes cannot be ignored.

### 3.4. Tubular Epithelial Cells

Tubular epithelial cells (TECs) are easily affected by metabolic disorders, inflammatory states, and hemodynamic changes. The expression of various cytokines, adhesion molecules, and extracellular matrix has attracted much attention in promoting renal fibrosis. The abnormality of renal tubular function in diabetic patients may precede or at least accompany pathological glomerular damage, which is characterized by hypertrophy of tubular epithelial cells, EMT, and glycogen accumulation of renal tubules [101]. Hyperglycemia affects epithelial cells of the distal tubule and proximal tubule differently. For example, distal tubular epithelial cells undergo Fas/Fas-L-mediated cell death in hyperglycemia, causing renal epithelial cell deficiency [102]. Chronic hyperglycemia promotes a pro-inflammatory response, or oxidative stress, in proximal tubular epithelial cells [103]. Among them, EMT is considered to be of great significance in renal tubulointerstitial fibrosis. It has been suggested that TECs acquire migration characteristics under appropriate stimulation and progress from tubular structures to newborn myofibroblasts [104]. Myofibroblasts are the key step of renal tubulointerstitial fibrosis. EMT is the transition process from polarized epithelial cells to a mesenchymal phenotype, including enhanced migration, invasion, apoptosis resistance, and increased ECM components, which is a potential source of fibroblasts [105]. However, it does not mean that TECs are the only source of myofibroblasts, such as vascular smooth muscle cells, endothelial cells, and most recently, pericytes are considered to be important sources of myofibroblasts [106,107]. 

Hyperglycemia also affects TECs. Still, surprisingly, the degree and specific mechanism of TECs phenotypic transformation induced by hyperglycemia is not clear. It has been reported that once the concentration of glucose in renal epithelial cells is higher than the interstitial glucose level; it will spread to the interstitium through a specific glucose transporter (GLUTs), in a high glucose environment, GLUTs are the cause of glucose flow into various glomerular cells, and glucose transport may play a central role in glomerulosclerosis [108,109]. In addition, glucose can induce cytokines in the kidney. TGF-β1 is an inducer of inflammation and fibrosis in the DN [110]. Some scholars say that when cells are exposed to glucose, the change in interstitial glucose concentration may regulate the synthesis of fibroblast transforming growth factor-β1 (TGF-β1), thus affecting the progression of interstitial fibrosis [111]. The researchers also found that TEC’s exposure to high glucose levels increased the expression of macrophage inflammatory protein-3α (MIP-3α) through TGF-β1-dependent pathway. MIP-3α is a chemokine ligand that can recruit memory T lymphocytes and induce inflammation. High glucose can also promote connective tissue growth factor (CTGF) expression and collagen accumulation in TECs. CTGF and deposited collagen play an essential role in renal interstitial fibrosis mediated by the hyperglycemia [112,113]. In vitro experiments show that the AGEs receptor induces epithelial-mesenchymal transformation through TGF β-dependent pathway, indicating that AGEs are involved in the process of renal fibrosis [114]. Although the renal interstitial fibrosis induced by EMT in TECs under high glucose stimulation plays a positive role in the pathological process of DN, it shows that the effect of TECs on renal cells induced by high glucose is very one-sided only in EMT. It has also been reported that high glucose induces dichlorofluorescein (DCF) sensitive, reactive oxygen species (ROS), ROS-activated signal transduction cascades, and transcription factors in TECs in a time-dependent manner, resulting in the up-regulation of genes and proteins involved in glomerular Mesangial dilatation and tubulointerstitial fibrosis, which also aggravates DN [115].
life-13-00539-t001_Table 1Table 1A summary of the injury mechanisms in four renal cells affected by high glucose and related supplements. MMP, matrix metalloproteinase; CTGF, connective tissue growth factor; Ang II, circulating angiotensin II; PDGF-B, platelet-derived growth factor-B; TGF-β1, trans-forming growth factor; ROS, reactive oxygen species; Wnt/β-catenin, beta-catenin dependent Wnt signaling; LC3-II, microtubule-associated protein light chain 3; APA, aminopeptidase A; AT2, angiotensin type 2 receptor; RAS, renin angiotensin system; TLR9, toll-like receptor9 pathway; PKC, protein kinase C; DAG, diacylglycerol; IGF-1, insulin-like growth factor; HIF-1α, hypoxia-inducible factor-1α; ET-1, endothelin-1; NO, nitric oxide; VEGF, vascular endothelial growth factor; EDN1/EDNRA, endothelin-1/endothelin-1 receptor type A; VEGFR2, vascular endothelial growth factor receptor 2; TGFβ, transforming growth factor-beta; IL-1, Interleukin 1; TRIM27, Tripartite motif-containing 27; SD protein, slit-diaphragm protein; PI3K/AKT, phosphatidylinositol 3-kinase/protein kinasesB; α-SMA, alpha smooth muscle actin; GSK3, glycogen synthase kinase-3; MEF2C, myocyte-specific enhancer factor 2C; MYF5, myogenic factor 5; PGC-1α, peroxisome proliferator-activated receptor-γ coactivator 1α; CARM1, coactivator-associated arginine methyltransferase 1; AMPKα, adenosine monophosphate (AMP)-activated protein kinase alpha; CB1R, cannabinoid receptor 1; p38MAPK, p38 mitogen activated protein kinase; LC3-ΙΙ, Microtubule-associated protein light chain 3-II; TRPC6, transient receptor potential channel C6; mTORC1, mechanistic target of rapamycin(mTOR) complex 1; ZO-1, zonula occludens-1; GLUTs, glucose transporter; TGF-β1, trans-forming growth factor; MIP-3α, macrophage inflammatory pro-tein-3α; DCF, dichlorofluorescein; TSG-6, tumor necrosis factor-stimulated gene 6; TRAF3IP2, TRAF3-interacting protein 2; NF-κB, nuclear factor kappa-light-chain-enhancer of activated B cells; RECK, reversion inducing cysteine rich protein with Kazal motifs.Cell NameConsequencesHG Damage MechanismReferenceMCMesangial matrix accumulationMMPs Activity↓, CTGF↑, Ang ΙΙ↑[70,71,116]ECMCollagen↑, Laminin↑, Fibrin↑[63]MC proliferationPDGF-B↑, TGF-β1↑, ROS↑[64,73]MC hypertrophyCell cycle arrest[65]MC apoptosisWnt/β-catenin↑, TGF-β1↑[66,67,68]GlomerulosclerosisTGF-βII receptor↑[69]Autophagy activationLC3-ΙΙ↑[116]MC degradationAPA↓, AT2receptor↑, RAS↑[117]Increased matrix accumulationCollagenase activity↓, Angiotensin II↑[118]Mitochondrial metabolism impairedCirculating mitochondrial DNA↑, TLR9 pathway activated↑[119]Hemodynamic abnormalityPKCα↑, PKC1↑, DAG↑[120]Production of fibronectin matrixIGF-1↑, PKC↑,[121]Inhibition of MC proliferationHeparan sulfate proteoglycans↑[122]MC bioenergy deficitO_2_ consumption rate↓, Extracellular acidification rate↓[123]GECsRenal fibrosisGECs to mesenchymal transition, The thickness of the endothelial glycocalyx↓, Mitochondrial ROS↑[79,124]EMTHIF-1α↑, ET-1↑[83]Mitochondrial dysfunctionET-1 receptor type A↑[85]HypertensionNO↓[86]Albumin permeability increasedGlycocalyx↓, Glycosaminoglycan↓[87]GECs dysfunctionVEGF-A, EDN1/EDNRA signaling↑[125,126]Deteriorate GECs damageVEGFR2↑, ROS↑, O_2_^−^↑[127]GECsEndothelial cell apoptosisTGFβ signal↑[125,128]Increased permeability of endothelial cellsIL-1↑, Hyaluronan↑[129]GECs dysfunctionPlatelet microparticles↑[130]GECs injuryTRIM27↑[131]PodocyteUrine proteinSD protein↓, Podocyte shedding↑[89]EMTNephrin↓, P-cadherin↓, Zonula occludens-1↓, PI3K/AKT↑, α-SMA↑, Desmin protein↑[90,91,94]Podocyte depletionGSK3↓, Snail protein↑, P-cadherin↓, Nephrin↓[92]Podocyte energy conversion disorderMEF2C↓, MYF5↓, PGC-1α↓[96]Podocyte apoptosisCARM1-AMPKα-Notch1-CB1R↑, CARM1↓, ROS↑, p38MAPK signaling↑[97,132]Decreased autophagy activityLC3-ΙΙ↑, beclin-1↑, Autophagosomes↑[133,134]Podocyte lossVEGF↓, TGFβ signal↑[135,136]Disruption of the slit diaphragm (SD)TRPC6↑[137]Focal detachment of podocytes from the GBMα3β1 integrin↓[138]Enhanced ER stressmTORC1↑[139]Podocyte insulin sensitivity decreasedNephrin↓[140]Podocyte foot reorganizationTRPC6↑, ZO-1↑[141]Podocyte and mesangial cell hypertrophy resulting in loss of adhesion to the GBMGlycemia↑[142]Lipid accumulation in podocytesVEGF-B↑, Fatty acid transport↑[143]TECsGlomerulosclerosisGLUTs↑[108,109]T lymphocyte accumulation, InflammationTGF-β1↑, MIP-3α↑[110]TECsRenal interstitial fibrosisCTGF↑, Collagen↑[112,113]Glomerular mesangial expansionDCF↑, ROS↑[115]Decreased autophagy activityMitochondrial fragmentation↑, LC3-II↓[144]Increase glomerular cellularityHyaluronan↑[145]Stimulate ANG gene expressionDAG↑, PKC↑[146]Thickening of renal tubular basement membraneFibronectin accumulation↑, Polyol pathway activation↑[147]Promoting fibrosisTSG-6 mRNA↑, Plasmin inhibitory activity↑[148]Inhibition of CK (reverse-induced cysteine-rich protein with Kazal motif) expressionTRAF3IP2↑, NF-κB and p38 MAPK activation↑, MMP2 activation↑, RECK↓[103]Note. The symbols “↑” means an increase and “↓” means decrease.

## 4. Summary

Diabetes-related kidney failure is caused by high blood sugar levels, and approximately one-third of patients with diabetes mellitus develop DN. The strict control of blood glucose in the early stage of diabetes can significantly reduce the incidence of diabetic microvascular and macroangiopathy. This review presents an update on the key mediators of hyperglycemia-induced renal cell injury and associated cellular mechanisms in DN. However, many studies have shown that the signaling mechanisms involved in this process and crosstalk between various cells still need to be further explored. Recognizing the principles of the effects of hyperglycemia on the renal cells in DN can significantly improve our grasp and understanding of glomerular and tubular pathological processes at the cellular level. It will help in the development of symptomatic treatments or drugs based on targets in renal disease, thus reducing the number of DN patients requiring dialysis.

## Figures and Tables

**Figure 1 life-13-00539-f001:**
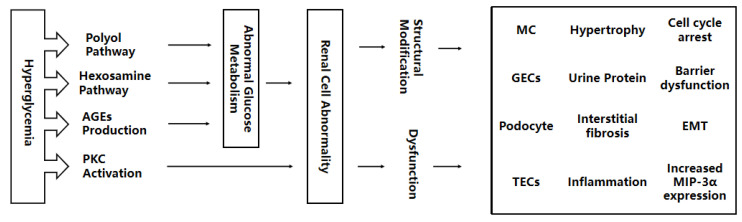
Hyperglycemia interferes with the activation of the polyol pathway, hexosamine pathway, AGEs production, and PKC, leading to abnormal glucose metabolism. Abnormal glucose metabolism accelerates renal structural modification and dysfunction, as well as kidney damage to MCs, GECs, adipocytes, and TECs. AGEs, advanced glycation end products; PKC, protein kinase C; MC, mesangial cell; GECs, glomerular endothelial cells; TECs, tubular epithelial cells; EMT, endothelial-interstitial transformation; MIP-3α, macrophage inflammatory protein-3α.

**Figure 2 life-13-00539-f002:**
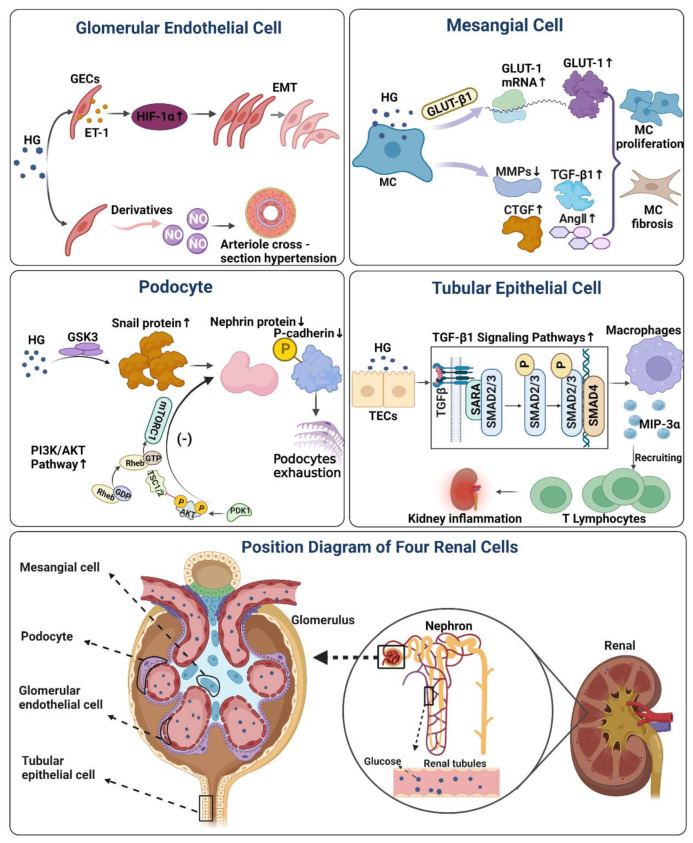
The effect and mechanism of high glucose on four renal cells in DN. Mesangial Cell Figure: Under the stimulation of a high glucose environment, it will induce MC to produce a large amount of GLUT-1 protein, while high sugar will affect the activity of MMPs, and induce the production of TGF-β1, Ang II, and CTGF. Finally, it will aggravate MC proliferation, sclerosis, and accelerate renal lesions. Glomerular Endothelial Cells figure: Hyperglycemia induces ET-1 production in GECs, and ET-1 mediates high glucose-induced EMT in GECs under the regulation of HIF-1α. glucose-mediated NO loss can directly cause arteriole cross-section hypertension and aggravate renal microvascular injury. Podocyte Figure: Reduced GSK3 activity in damaged podocytes leads to increased expression of Snail in podocytes, and upregulated Snail suppresses protein expression of P-calmodulin and nephrin, finally causing podocyte depletion. Activation of PI3K/AKT signaling pathway in podocytes by high glucose likewise inhibits protein expression of podocalyxin and nephrin, which on also induces podocyte phenotypic transformation. Tubular Epithelial Cells Figure: Exposure of TECs to HG levels increased macrophage inflammatory protein-3α (MIP-3α) expression through a TGF-β1-dependent pathway, recruiting memory T lymphocytes and inducing renal inflammation. HG, hyperglycemia; GLUT-1, glucose transporter-1; MMPs, matrix metalloproteinases; TGF-β1, transforming growth factor; CTGF, connective tissue growth factor; AngII, circulating angiotensin II; HIF-1α, hypoxia-inducible factor-1α; ET-1, endothelin-1; EMT, endothelial-interstitial transformation; GSK3, glycogen synthase kinase-3; PDK1, 3-phosphoinositide-dependent protein kinase-1; AKT, protein kinases B; mTOR, mammalian target of rapamycin; TSC1/2, tuberous sclerosis1/2; MIP-3α, macrophage inflammatory protein-3α; (−), negative Feedback; ↑, up-regulation; ↓, down-regulation.

## Data Availability

Not applicable.

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
