# Peer review of "The Mechanism of Hyperglycemia-Induced Renal Cell Injury in Diabetic Nephropathy Disease: An Update"

_life, 2023, doi:10.3390/life13020539_

Round 1

Reviewer 1 Report (Previous Reviewer 1)

Most of my suggestions and criticisms have been addressed appropriately. However, some details introduced in the revisions need attention:

Line 52: insufficient insulin secretion reduces the ability to store glucose also or in the first place in liver. 

Line 89: What is meant by increased liver glucose transport? Increased glucose output by the liver?

Lines 119-120: please remove "or accumulation".

Lines 199-202: I guess pentosidine is reabsorbed FROM the renal tubules. When pentosidine is reabsorbed, modified or degraded in the renal tubular cells, how can it that ultimately be removed in the urine? Reduced renal clearance is the main cause of WHICH phenomenon?

Lines 244-256: what is PCK?

Line 282: "on"?

Author Response

Reviewer 2 Report (New Reviewer)

Authors defined the causes and consequences of renal damage brought on by hyperglycemia in this review article. Although the review article is very well written and explained, it still needs a few minor edits before it can be accepted for publication

This review article does not pay enough attention to what is crucial and instead covers a vast array of irrelevant information. The authors must summarize or remove unnecessary information.

 The article needs extensive English editing .

Author Response

Reviewer 3 Report (New Reviewer)

1. The format of table 1 (vertical box) needs revision.

2. Currently, RAAS system blockage is the main therapy for DM nephropathy clinically. Is there new progress about the Pathological roles of RAAS activation in DM nephropathy?

Author Response

please see attached.

This manuscript is a resubmission of an earlier submission. The following is a list of the peer review reports and author responses from that submission.

Round 1

Reviewer 1 Report

In this paper, the authors review progress in our understanding of the mechanisms by which hyperglycemia might induce renal cell injury in the development of diabetic kidney disease (DKD). The authors first describe four well-known pathways of hyperglycemia-induced cell injury, i.e. the polyol, hexosamine, AGEs and PKC pathway. Next, the hyperglycemia-induced injury of four cell types are summarized, i.e. the glomerular mesangial cells, the glomerular endothelial cells, the podocytes and the tubular epithelial cells. Although a review on this subject is potentially interesting, the paper is hard to read due to many repetitions, poor language, sloppiness and mistakes.

Although stated in lines 26/27 it is not made clear why the kidney is the most vulnerable organ. Similarly, why is the kidney the first organ to be affected by the accumulation of AGEs (line 192)?

Epithelial cells from the distal tubules and loop of Henle are distinct from the proximal tubules, and may be affected differently by hyperglycemia. In addition, it might have been interesting to include possible effects of hyperglycemia on the juxtaglomerular cells.

Figure 1: What are TELs? What are Sertoli cells in the kidney?

Figure 2: there are not supposed to be erythrocytes inside renal tubules. The four panels referring to the different mechanisms in the four specified cell types are too small to be read properly.

Table 2 should have been table 1, and should have an appropriate title.

Line 8: microproteinuria

Line 9: diabetes is also characterized by elevated fatty acid levels, why is chronic hyperglycemia the most important pathogenic factor of DKD?

Line 27: pro-inflammatory fibers??

Line 34: globules??

Line 39: there is no such thing as microalbumin

Line 45-47: this sentence does not make sense.

Line 58-59: most proteins are enzymatically glycosylated, but HbA1c is not. HbA1 is glycated, not glycosylated. All kinds of proteins may be glycated non-enzymatically, HbA1c is no exception.

Line 64: large amounts of proteinuria?

Line 83-84: … resulting in ….reduced treatment?

From here, I give up commenting.